# Acute Kidney Injury in Pediatric Patients on Extracorporeal Membrane Oxygenation: A Systematic Review and Meta-analysis

**DOI:** 10.3390/medicines6040109

**Published:** 2019-11-01

**Authors:** Panupong Hansrivijit, Ploypin Lertjitbanjong, Charat Thongprayoon, Wisit Cheungpasitporn, Narothama Reddy Aeddula, Sohail Abdul Salim, Api Chewcharat, Kanramon Watthanasuntorn, Narat Srivali, Michael A. Mao, Patompong Ungprasert, Karn Wijarnpreecha, Wisit Kaewput, Tarun Bathini

**Affiliations:** 1Department of Internal Medicine, University of Pittsburgh Medical Center Pinnacle, Harrisburg, PA 17105, USA; hansrivijitp@upmc.edu; 2Department of Internal Medicine, Bassett Medical Center, Cooperstown, NY 13326, USA; ploypinlert@gmail.com (P.L.); kanramon@gmail.com (K.W.); 3Division of Nephrology and Hypertension, Mayo Clinic, Rochester, MN 55905, USA; api.che@hotmail.com; 4Division of Nephrology, University of Mississippi Medical Center, Jackson, MS 39216, USA; sohail3553@gmail.com; 5Department of Internal Medicine, Deaconess Health System, Evansville, IN 47747, USA; dr.anreddy@gmail.com; 6Division of Pulmonary and Critical Care Medicine, St. Agnes Hospital, Baltimore, MD 21229, USA; nsrivali@gmail.com; 7Department of Medicine, Mayo Clinic, Jacksonville, FL 32224, USA; mao.michael@mayo.edu (M.A.M.); dr.karn.wi@gmail.com (K.W.); 8Department of Rheumatic and Immunologic Diseases, Cleveland Clinic, Cleveland Clinic Lerner College of Medicine of Case Western Reserve University, Cleveland, OH 44106, USA; p.ungprasert@gmail.com; 9Department of Military and Community Medicine, Phramongkutklao College of Medicine, Bangkok 10400, Thailand; wisitnephro@gmail.com; 10Department of Internal Medicine, University of Arizona, Tucson, AZ 85721, USA; tarunjacobb@gmail.com

**Keywords:** AKI, ECMO, extracorporeal membrane oxygenation, incidence, mortality

## Abstract

**Background:** Acute kidney injury (AKI) is a well-established complication of extra-corporal membrane oxygenation (ECMO) in the adult population. The data in the pediatric and neonatal population is still limited. Moreover, the mortality risk of AKI among pediatric patients requiring ECMO remains unclear. Thus, this meta-analysis aims to assess the incidence of AKI, AKI requiring renal replacement therapy and AKI associated mortality in pediatric/neonatal patients requiring ECMO. **Methods:** A literature search was performed utilizing MEDLINE, EMBASE, and the Cochrane Database from inception through June 2019. We included studies that evaluated the incidence of AKI, severe AKI requiring renal replacement therapy (RRT) and the risk of mortality among pediatric patients on ECMO with AKI. Random-effects meta-analysis was used to calculate the pooled incidence of AKI and the odds ratios (OR) for mortality. **Results:** 13 studies with 3523 pediatric patients on ECMO were identified. Pooled incidence of AKI and AKI requiring RRT were 61.9% (95% confidence interval (CI): 39.0–80.4%) and 40.9% (95%CI: 31.2–51.4%), respectively. A meta-analysis limited to studies with standard AKI definitions showed a pooled estimated AKI incidence of 69.2% (95%CI: 59.7–77.3%). Compared with patients without AKI, those with AKI and AKI requiring RRT while on ECMO were associated with increased hospital mortality ORs of 1.70 (95% CI, 1.38–2.10) and 3.64 (95% CI: 2.02–6.55), respectively. **Conclusions:** The estimated incidence of AKI and severe AKI requiring RRT in pediatric patients receiving ECMO are high at 61.9% and 40.9%, respectively. AKI among pediatric patients on ECMO is significantly associated with reduced patient survival.

## 1. Introduction

For the past 20 years, extra-corporal membrane oxygenation (ECMO) use has remarkably increased, as it has become an essential rescue therapy for severe refractory cardiac and pulmonary dysfunction in both adults and children [1]. In general, ECMO provides temporary support for patients who have a predicted mortality rate of 80% and above [2]. Indications for ECMO can be categorized into: (1) Respiratory Support and (2) Cardiac Support. Based on the indication for ECMO, the modality of ECMO is adapted for the patient. Veno-venous (VV) ECMO provides only respiratory support, and requires patients to have stable hemodynamics, whereas veno-arterial (VA) ECMO provides dual cardiac and respiratory support [1,2,3,4,5,6,7]. In VA ECMO, the blood bypasses both the heart and lungs. Generally, there is minimal vascular pulsatility in VA ECMO, while there is considerable pulsatility in VV ECMO. Pulsatility has been suggested to be essential in maintaining optimal perfusion to the vital organs, including the brain, heart and kidneys [8]. For instance, Pappalardo et al. [9] has shown that the incidence of acute ischemic stroke in VV ECMO is 1.4% compared to 3.8% in VA ECMO [10].

A recent meta-analysis has demonstrated that the pooled incidence of AKI and severe AKI requiring renal replacement therapy (RRT) among adult patients on ECMO was 63% and 45%, respectively. Furthermore, adult ECMO patients with AKI requiring RRT have a significantly increased mortality with a pooled odds ratio (OR) of 3.73 [11]. Although the incidence and associated mortality of AKI in adults on ECMO are widely reported, comparable data in the pediatric population are limited. While one study has reported a mortality rate of 27.4 and 41.6%, respectively, for neonatal and pediatric patients who received ECMO support for non-cardiac indications [12], the impact of AKI and its severity on outcomes is less clear. Furthermore, the need for RRT during ECMO is considered an independent risk factor for the failure to wean from ECMO [13]. Therefore, we performed this meta-analysis to assess the incidence rate of AKI and its associated mortality in pediatric patients receiving ECMO for various indications. The results of our study will help define the burden of AKI and its clinical impact on pediatric patients receiving ECMO, and allow comparisons to the adult counterpart.

## 2. Method

### 2.1. Literature Review and Search Strategy

We conducted a systematic literature review of Ovid MEDLINE, EMBASE and the Cochrane Database of Systematic Reviews until June 2019 to assess the incidence and mortality risk of AKI in pediatric patients on ECMO. Independent reviewers (C.T. and P.L.) conducted a systematic literature search using a search strategy that incorporated the search terms “extracorporeal membrane oxygenation” OR “ECMO” AND “acute kidney injury” OR “acute renal failure”, as shown in Appendix A. A manual search for potentially relevant studies utilizing the references of the initial included articles was subsequently performed. There was no language limitation. This systematic review was conducted following the PRISMA (Preferred Reporting Items for Systematic Reviews and Meta-Analysis) [14].

### 2.2. Selection Criteria

Inclusion criteria consisted of observational studies or clinical trials that evaluated both the incidence of acute kidney injury (AKI) and/or severe AKI requiring RRT and mortality risk of AKI in pediatric patients (aged younger than 18 years old) on ECMO. Retrieved articles were individually assessed for study eligibility by the two investigators previously noted. Any discrepancies were resolved by mutual consensus.

### 2.3. Data Abstraction

A data collecting form was utilized to derive the following information from the individual studies: Title, study year, publication year, name of authors, country where the study was performed, ECMO type, definition of AKI, incidence rate of AKI, incidence rate of AKI requiring RRT and risk of mortality in pediatric patients on ECMO with AKI.

### 2.4. Statistical Analysis

Analyses were performed using the Comprehensive Meta-Analysis 3.3 software (version 3; Biostat Inc, Englewood, NJ, USA). Pooled AKI incidence and mortality risk of included studies were incorporated by the generic inverse variance method of DerSimonian-Laird, which indicated the weight of each study depending on its variance [15]. Because of the likelihood of inter-observation variance, we utilized a random-effects model for meta-analyses of the incidence and mortality risk of AKI among pediatric patients receiving ECMO. Statistical heterogeneity of studies was evaluated by the Cochran’s Q test (statistically significant as *p* < 0.05) and the *I*^2^ statistic (≤25% represents insignificant heterogeneity, 26% to 50% represents low heterogeneity, 51% to 75% represents moderate heterogeneity, and ≥75% represents high heterogeneity) [16]. Publication bias was evaluated via the Egger test [17].

## 3. Results

Our search strategy produced a total of 1741 potentially eligible articles. 887 articles were excluded because they were duplicated. and 684 articles because they were case reports, correspondences, review articles, in vitro studies, or animal studies. 170 articles remained for full-length review. Forty one articles were subsequently excluded because the study cohort was an adult patient population, and 116 studies were excluded due to a lack of data on the outcomes of interest. Consequently, 13 cohort studies [12,18,19,20,21,22,23,24,25,26,27,28,29] comprised of 3523 pediatric patients on ECMO were identified in this systematic review. The systematic review process is presented in Figure 1. The characteristics of the included studies are demonstrated in Table 1. The kappa coefficient (0.87) indicated that agreement between the authors was acceptable.

### 3.1. Incidence of AKI among Pediatric Patients on ECMO

Pooled estimated incidence of AKI and severe AKI requiring RRT were 61.9% (95%CI: 39.0–80.4%, *I*^2^ = 98%, Figure 2A) and 40.9% (95%CI: 31.2–51.4%, *I*^2^ = 86%, Figure 2B), respectively. We additionally performed a meta-analysis limited to studies using standard AKI definitions; this showed a pooled estimated AKI incidence of 69.2% (95%CI: 59.7–77.3%, *I*^2^ = 83%, Appendix A).

Subgroup analysis based on patient population (cardiac surgery vs. non-cardiac surgery) was performed. Pooled incidence of AKI was 81.4% (95%CI: 54.4–94.2%, *I*^2^ = 84%, Appendix A) among patients after cardiac surgery and 52.3% (95%CI: 27.3–76.2%, *I*^2^ = 99%, Appendix A) among non-cardiac surgery patients. Pooled incidence of AKI requiring RRT was 52.2% (95%CI: 32.5–71.2%, *I*^2^ = 88%, Appendix A) among patients after cardiac surgery and 30.2% (95%CI: 18.7–44.9%, *I*^2^ = 84%, Appendix A) among non-cardiac surgery patients.

### 3.2. Mortality in Pediatric ECMO Patients with AKI 

Table 1 and Table 2 demonstrate the AKI-associated mortality rate and risk in pediatric patients on ECMO, respectively. Pooled estimated hospital or 90-day mortality rates of pediatric ECMO patients with AKI and AKI requiring RRT were 51.7% (95%CI: 26.5–76.0%, *I*^2^ = 96%, Appendix A) and 61.8% (95%CI: 48.8–73.4%, *I*^2^ = 74%, Appendix A), respectively.

The hospital mortality pooled OR in pediatric ECMO patients with AKI and severe AKI requiring RRT were 1.70 (95% CI: 1.38–2.10, *I*^2^ = 0%, Figure 3A) and 3.64 (95% CI, 2.02–6.55, *I*^2^ = 58%, Figure 3B), respectively. Meta-regression analysis was additionally performed, and it demonstrated that year of study was not correlated with either mortality risk associated with AKI (*p* = 0.65) or AKI requiring RRT (*p* = 0.46). 

### 3.3. Assessment for Publication Bias

Funnel plots (Appendix A) and Egger’s regression asymmetry tests were used to evaluate for publication bias. We found no significant publication bias for the incidence of AKI and severe AKI requiring RRT (*p* = 0.16 and *p* = 0.74, respectively).

## 4. Discussion

In this study, we have shown that the incidence of AKI and severe AKI in the pediatric population requiring ECMO is high (62% and 41%, respectively), especially in cardiac surgery patients requiring ECMO. AKI is associated with high mortality (OR 3.64) and AKI-associated mortality did not change over time. Similar to the adult population, our study in pediatric patients concurs that ECMO is associated with a higher risk of AKI and hospital mortality [11].

The mechanism by which ECMO support predisposes patients to AKI is not well established. Certainly, patients requiring ECMO often possess many known comorbidities and risk factors for AKI, such as low cardiac output, septic shock, anemia, diabetes, exposure to nephrotoxic drugs, etc. [30,31]. In general, there are fewer comorbidities among pediatric patients. Conversely, their renal reserve may not have fully developed [12].

However, it is proposed that ECMO itself is capable of inducing AKI via detrimental (1) hemodynamic alterations; (2) hormonal imbalances; and (3) systemic inflammatory cytokines [11,30,31]. Hemodynamic alterations may include the potential loss of pulsatility with VA ECMO versus the maintenance of pulsatility with VV ECMO. Evidence is supportive of the vital impact of pulsatility on maintaining renal cortical perfusion, thus helping to prevent acute tubular necrosis [8]. Slight hemodynamic changes could potentiate a significant reduction in renal perfusion. Current data on ECMO-associated hormonal changes show conflicting evidence. Saito et al. hypothesized that upregulation of plasma renin activity may be an adaptive response to the loss of pulsatile renal perfusion [32]. However, Semmekrot et al. observed a reduction of atrial natriuretic peptide (ANP) level, plasma renin activity (PRA) and angiotensin II levels in neonates requiring VA ECMO [33]. Further clinical investigation is needed to better understand the impact and interaction of different hormones on renal perfusion. Finally, studies have demonstrated that the continuous exposure of blood to the non-endothelialized/non-biological ECMO interface results in an activation of proinflammatory cytokines, such as tumor necrosis factor-alpha (TNF-α), IL (interleukin)-1β, IL-6, IL-8 [34,35].

Our study shows that the incidence of AKI, severe AKI requiring RRT and AKI-associated hospital mortality between adults and pediatric patients are similar [11]. This further supports ECMO as a potential independent predictor for AKI. However, we found a significant higher incidence of AKI and RRT in pediatric patients that the indication for ECMO was after cardiac surgery, than those who required ECMO for non-cardiac surgery related conditions. Despite improved knowledge of the indications for ECMO, patient comorbidities, and associated risk factors that may contribute to ECMO-associated AKI, ECMO as an independent risk factor for AKI cannot be determined from this meta-analysis alone. A pooled meta-analysis of both adult and pediatric patients assessing the incidence of AKI and mortality following ECMO support would be useful. Furthermore, a subgroup analysis comparing adult and pediatric patients would further elaborate whether ECMO is associated with AKI and increased mortality.

There were a few limitations in this study. Most of the included studies defined AKI based on serum creatinine levels, while studies utilizing the more sensitive urine output criteria or use of novel AKI biomarkers were limited [36]. Moreover, our meta-analysis is primarily based on observational studies, making it impossible to establish a causal relationship. Future data from population-based studies or prospective clinical trials focusing on AKI prevention in ECMO patients would be beneficial.

## 5. Conclusions

Our study demonstrates that AKI is common following ECMO in pediatric patients with an incidence of 68%. Approximately 40% of pediatric patients on ECMO develop severe AKI requiring RRT. We have also shown that presence of AKI is associated with a higher risk of hospital mortality. There is no difference in mortality between recent and remote studies. Clinical trials focusing on preventive measures for AKI with ECMO therapy is also encouraged.

## Figures and Tables

**Figure 1 medicines-06-00109-f001:**
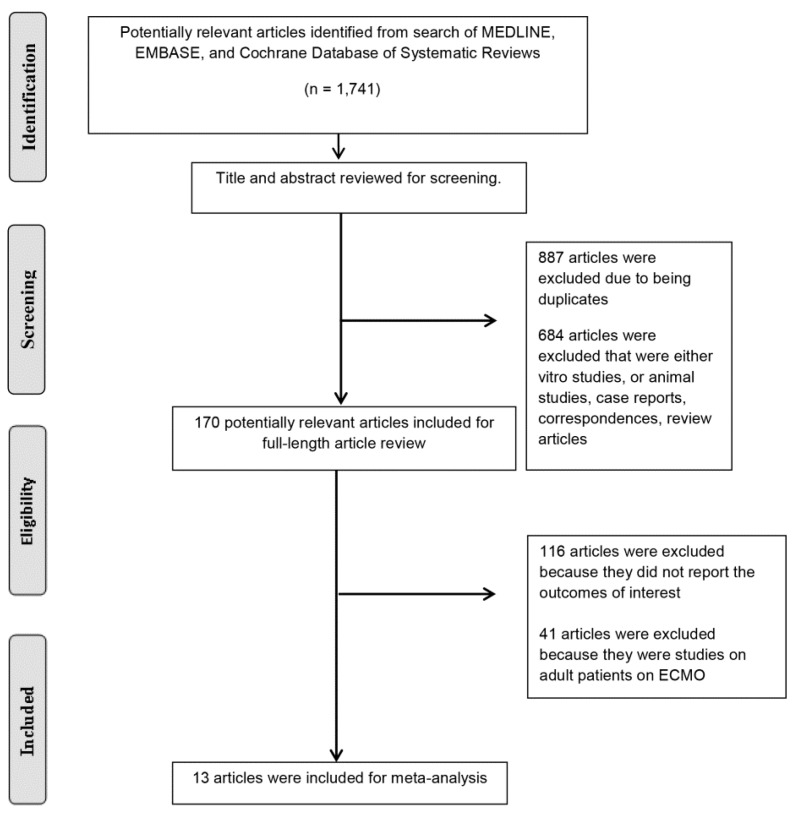
Flow diagram of our search strategy.

**Figure 2 medicines-06-00109-f002:**
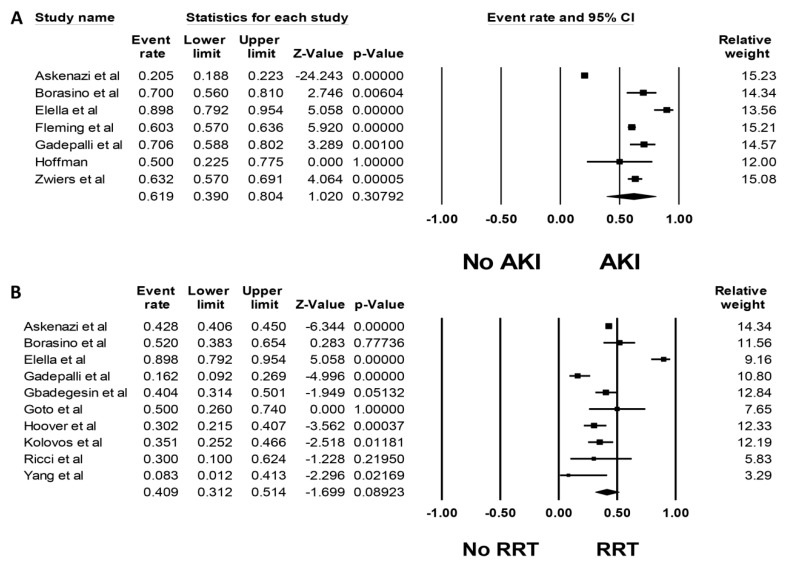
Forest plots of included studies on pediatric patients requiring extra-corporal membrane oxygenation (ECMO) that evaluated (**A**) incidence of AKI and (**B**) incidence of AKI requiring renal replacement therapy (RRT).

**Figure 3 medicines-06-00109-f003:**
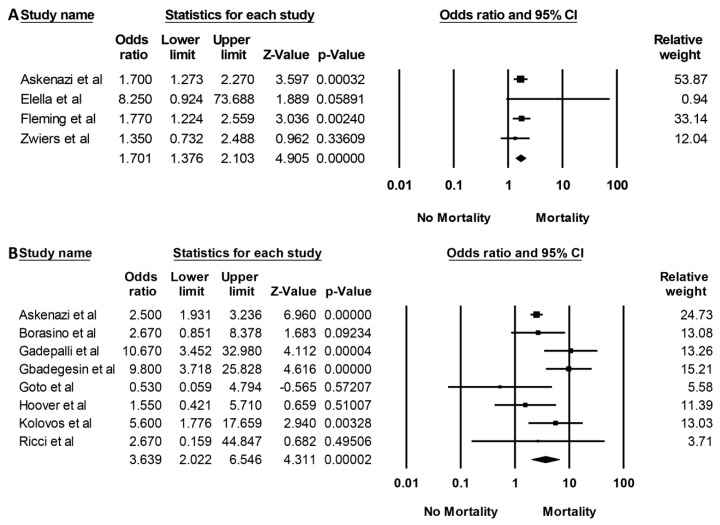
Forest plots of included studies on pediatric patients requiring ECMO that evaluated (**A**) hospital mortality with AKI and (**B**) hospital mortality with AKI necessitating RRT.

**Table 1 medicines-06-00109-t001:** Included studies in this systematic review of acute kidney injury (AKI) incidence and mortality in pediatric patients requiring ECMO.

Study	Year	Country	Patient Population	Number	Definition of AKI	Incidence of AKI	Mortality
Kolovos et al. [18]	2003	USA	Patients on ECMO within seven days after cardiac surgery	74	RRT	RRT 26/74 (35%)	Hospital mortality RRT 20/26 (77%)
Hoover et al. [19]	2008	USA	Patients aged 1 month to 18 years with respiratory failure requiring ECMO	86	CRRT	CRRT 26/86 (30%)	Hospital mortality RRT 7/26 (27%)
Gbadegesin et al. [20]	2009	USA	Patients aged < 3 years requiring ECMO after cardiac surgery	104	CRRT	CRRT 42/104 (40%)	Hospital mortality RRT 35/42 (83%)
Gadepalli et al. [21]	2010	USA	Congenital diaphragmatic hernia patients requiring ECMO	68	AKI: RIFLE	AKI 48/68 (71%) RIFLE—failure AKI 33/68 (49%) CRRT 11/68 = 16%	Mortality RIFLE—failure AKI 24/33 (73%)
Goto et al. [22]	2011	USA	Patients aged 19 days to 20 years with respiratory failure and/or heart failure requiring ECMO	14	RRT	RRT 7/14 (50%)	Hospital mortality RRT 2/7 (29%)
Askenazi et al. [12]	2011	USA	All non-cardiac patients requiring ECMO	9903 Neonates -7941 Children -1962	SCr > 1.5 mg/dL or ICD-9 for acute renal failure	Neonates AKI—638/7941 (8%) RRT—1786/7941 (22%) Children AKI—402/1962 (20%) RRT—840/1962 (43%)	Neonates Hospital mortality AKI—413/638 (65%) RRT—863/1786 (48%) Children Hospital mortality AKI—264/402 (66%) RRT—487/840 (58%)
Ricci et al. [23]	2012	Italy	Patients aged 13 days to 13 years on VA ECMO after cardiac surgery	10	CRRT	CRRT 3/10 (30%)	Hospital mortality RRT 2/3 (66%)
Hoffman et al. [24]	2013	USA	Patients with persistent hypoxia or cardiovascular instability requiring ECMO	10	AKI; (1) urine output < 1 ml/kg/h with SCr > 1 mg/dL for 24 hours, (2) SCr of > 1.5 mg/dL, (3) failure to improve creatinine clearance by > 50%	AKI 5/10 (50%)	N/A
Zwiers et al. [25]	2013	Netherlands	Neonates aged < 28 days requiring ECMO	242	AKI: RIFLE	AKI 153/242 (63%)	Hospital mortality AKI 43/153 (28%)
Fleming et al. [26]	2016	USA	Pediatric patients aged < 18 requiring ECMO	832	AKI: KDIGO	AKI by SCr—502/832 (60%) AKI by SCr + renal support therapy—615/832 (74%)	N/A
Yang et al. [27]	2016	China	Patients aged 1 to 13 years with refractory cardiopulmonary failure requiring ECMO	12	CRRT	CRRT 1/12 (8%)	Hospital mortality RRT 0/1 (0%)
Elella et al. [28]	2017	Saudi Arabia	Pediatric patients requiring VA-ECMO after cardiac surgery	59	AKI pRIFLE	AKI 53/59 (90%) RRT 29/59 (49%)	Hospital mortality AKI 33/53 (62%)
Borasino et al. [29]	2018	USA	Pediatric cardiac patient on ECMO in CICU	50	AKI: increase in SCr of 200% from baseline	AKI 35/50 (70%) RRT 26/50 (52%)	Hospital mortality RRT 16/26 (61%)

Abbreviations: AKIN, Acute Kidney Injury Network; COPD, chronic obstructive pulmonary disease; CRRT, continuous renal replacement therapy; KDIGO, Kidney Disease Improving Global Outcomes; pRIFLE, Pediatric Risk, Injury, Failure, Loss of kidney function; RIFLE, Risk, Injury, Failure, Loss of kidney function, and End-stage kidney disease; RRT, renal replacement therapy; SCr, serum creatinine; ECMO, extra-corporal membrane oxygenation.

**Table 2 medicines-06-00109-t002:** Mortality Risk of AKI among pediatric patients requiring ECMO.

Study	Mortality Rate	OR for Mortality
Kolovos et al. [18]	Hospital mortality RRT 20/26 (77%)	Hospital mortality RRT: OR 5.6 (1.8–17.9)
Hoover et al. [19]	Hospital mortality RRT 7/26 (27%)	Hospital mortality CRRT: 1.55 (0.42–5.70)
Gbadegesin et al. [20]	Hospital mortality RRT 35/42 (83%)	Hospital mortality CRRT: 9.8 (3.7–25.7)
Gadepalli et al. [21]	Mortality RIFLE—failure AKI 24/33 (73%)	Hospital mortality RIFLE—Failure AKI: 10.67 (3.45–32.96)
Goto et al. [22]	Hospital mortality RRT 2/7 (29%)	Hospital mortality RRT: 0.53 (0.06–4.91)
Askenazi et al. [12]	Neonates Hospital mortality AKI—413/638 (65%) RRT—863/1786 (48%) Children Hospital mortality AKI—264/402 (66%) RRT—487/840 (58%)	Hospital mortality Neonates AKI: 3.2 (2.6–4.0) RRT: 1.9 (1.6–2.2) Children AKI: 1.7 (1.3–2.3) RRT: 2.5 (1.9–3.2)
Ricci et al. [23]	Hospital mortality RRT 2/3 (66%)	Hospital mortality CRRT: 2.67 (0.16–45.14)
Zwiers et al. [25]	Hospital mortality AKI 43/153 (28%)	Hospital mortality AKI: 1.35 (0.73–2.48)
Fleming et al. [26]	N/A	Hospital mortality AKI-SCr: 1.77 (1.22–2.55) AKI-SCr+RST: 2.50 (1.61–3.90)
Yang et al. [27]	Hospital mortality RRT 0/1 (0%)	Hospital mortality CRRT: 0/1 vs. 4/11
Elella et al. [28]	Hospital mortality AKI 33/53 (62%)	Hospital mortality AKI: 8.25 (0.90–75.79)
Borasino et al. [29]	Hospital mortality RRT 16/26 (61%)	Hospital morality RRT: 2.67 (0.85–8.37)

Abbreviations: AKIN, Acute Kidney Injury Network; COPD, chronic obstructive pulmonary disease; CRRT, continuous renal replacement therapy; KDIGO, Kidney Disease Improving Global Outcomes; pRIFLE, Pediatric Risk, Injury, Failure, Loss of kidney function; RIFLE, Risk, Injury, Failure, Loss of kidney function, and End-stage kidney disease; RRT, renal replacement therapy; RST, renal support therapy; SCr, serum creatinine.

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
