# Peer review of "Acute Kidney Injury in Pediatric Patients on Extracorporeal Membrane Oxygenation: A Systematic Review and Meta-analysis"

_medicines, 2019, doi:10.3390/medicines6040109_

Round 1
Reviewer 1 Report
This work is about AKI and ECMO use in the pediatric patients. It is very good for us to understand the incidence of AKI and the influnence on mortality. This work let the readers know AKI is common following ECMO in pediatric patients with an 193 incidence of 68%. Up to 40% of pediatric patients who are on ECMO will eventually need RRT.
Author Response
We thank you for reviewing our manuscript and for your critical evaluation. We appreciated your nice comments regarding importance of findings from our study. We are confident that this manuscript will help future research on AKI in pediatric/neonatal patients requiring ECMO.

Reviewer 2 Report
The author performed a systematic review and meta-analysis on reported paediatric ECMO studies, focusing on the prevalence of AKI and RRT, and their association with patient outcome. The paper discovered that ECMO patients have significantly higher risk of developing AKI, and there is a strong correlation between AKI and patient survival.
The overall analysis appears to be sound, and as the authors alluded to in the discussion, AKI has already been associated with ECMO in the adult population. One key concern is the effect of cardiac surgery and its role on AKI. Again, mentioned in the discussion, unlike the adult study, most of the patients are cardiac surgery patients, which is known to have a high correlation with AKI. Therefore, it is difficult to determine if the high level of AKI incidence is due to ECMO, or in that patient group. Is there sufficient data to analyse them as two different group? At the very minimum, additional discussion on this will be helpful.
Some other comments:
Figure 2, as event rate cannot go beyond 0 and 1, unclear why the range -2 to 2 was used. Figure 2a, Askenazi has one of the largest cohort included in this review, why isn't it included in Figure 2A? Unclear why there are many p-value that is 1 or 0.000. May I ask how is relative weight determined in this study?Author Response
Response to Reviewer #2
The author performed a systematic review and meta-analysis on reported paediatric ECMO studies, focusing on the prevalence of AKI and RRT, and their association with patient outcome. The paper discovered that ECMO patients have significantly higher risk of developing AKI, and there is a strong correlation between AKI and patient survival.
The overall analysis appears to be sound, and as the authors alluded to in the discussion, AKI has already been associated with ECMO in the adult population.
Response: We thank you for reviewing our manuscript and for your critical evaluation.
Comment #1
One key concern is the effect of cardiac surgery and its role on AKI. Again, mentioned in the discussion, unlike the adult study, most of the patients are cardiac surgery patients, which are known to have a high correlation with AKI. Therefore, it is difficult to determine if the high level of AKI incidence is due to ECMO, or in that patient group. Is there sufficient data to analyse them as two different groups? At the very minimum, additional discussion on this will be helpful.
Response: The reviewer raised very important point. We appreciated this important comment. We additionally performed subgroup analysis based on cardiac surgery status as the reviewer’s suggestion. We found that those with cardiac surgery on ECMO developed more AKI. We have added this important finding in the result as well as discussion as the reviewer’s suggestion. The following text has been added.
“Subgroup analysis based on patient population (cardiac surgery vs. non-cardiac surgery) was performed. The pooled estimated incidence of AKI was 81.4% (95%CI: 54.4%-94.2%, I2 = 84%, Figure S2) among patients after cardiac surgery and 52.3% (95%CI: 27.3%-76.2%, I2 = 99%, Figure S2) among non-cardiac surgery patients. The pooled estimated incidence of AKI requiring RRT was 52.2% (95%CI: 32.5%-71.2%, I2 = 88%, Figure S3) among patients after cardiac surgery and 30.2% (95%CI: 18.7%-44.9%, I2 = 84%, Figure S3) among non-cardiac surgery patients.”
Comment #2
Figure 2, as event rate cannot go beyond 0 and 1, unclear why the range -2 to 2 was used.
Response: We appreciated the reviewer’s important comment. We agree with the reviewer’s comment and thus corrected our Forest plots to demonstrate event rate between 0 and 1.
Comment #3
Figure 2a, Askenazi has one of the largest cohorts included in this review, why isn't it included in Figure 2A?
Response: The reviewer is correct. We apologize for the error. Askenazi et al is now included in Figure 2A as the reviewer’s suggestion. We also updated the finding of study throughout revised manuscript as the reviewer’s suggestion.
Comment #4
Unclear why there are many p-value that is 1 or 0.000.
Response: We apologize for this inconvenience. It is limitation of comprehensive meta-analysis software version 3.3.070. We have increased decimals of p-value as we can to better describe the findings. However, comprehensive meta-analysis software cannot generate p<0.001 in the Forrest plots.
Comment #5
May I ask how is relative weight determined in this study?
Response: The reviewer asked very important question. We used random-effects model in our meta-analysis. Thus, under the random effects model these studies are drawn from a range of populations in which the effect size varies and our goal is to summarize
this range of effects. Each study is estimating an effect size for its unique population, and so each must be given appropriate weight in the analysis.
We greatly appreciated the reviewer’s time and comments to improve our manuscript.

Reviewer 3 Report
Excellent paper on review of data for AKI development for pediatric patients on ECMO.
Just minor suggestions:
For discussion, lines 161 to 163, please add reference to this sentence.
For table 2 - could delete if add the OR for mortality to Table 1.
For conclusion - overstating OD for mortality since that is AKI + RRT number. Please adjust this statement.
Thanks for opportunity to review.
Author Response
Response to Reviewer #3
Excellent paper on review of data for AKI development for pediatric patients on ECMO.
Response: We thank you for reviewing our manuscript and for your critical evaluation.
Comment #1
For discussion, lines 161 to 163, please add reference to this sentence.
Response: We appreciated the reviewer’s important comment. We agree and the references have been added as suggested.
Comment #2
For table 2 - could delete if add the OR for mortality to Table 1.
Response: We appreciated the reviewer’s comments. If it is possible, we would like to keep table 2 to emphasize the importance and impact of AKI-associated mortality.
Comment #3
For conclusion - overstating OR for mortality since that is AKI + RRT number. Please adjust this statement.
Response: We appreciated the reviewer’s important comment. We agree and we have revised our conclusion as the reviewer’s suggestion. The following text has been revised in the conclusion.
“Our study demonstrates that AKI is common following ECMO in pediatric patients with an incidence of 68%. Approximately 40% of pediatric patients on ECMO develop severe AKI requiring RRT. We have also shown that presence of AKI is associated with higher risk of hospital mortality. There is no difference in mortality between recent and remote studies.”
We greatly appreciated the reviewer’s time and comments to improve our manuscript.

Round 2
Reviewer 2 Report
I am happy with the authors' responses.